# Risk Factors for Calcium-Phosphate Disorders after Thyroid Surgery

**DOI:** 10.3390/biomedicines11082299

**Published:** 2023-08-18

**Authors:** Monika Sępek, Dominik Marciniak, Mateusz Głód, Krzysztof Kaliszewski, Jerzy Rudnicki, Beata Wojtczak

**Affiliations:** 1Department of General, Minimally Invasive and Endocrine Surgery, Wroclaw Medical University, Borowska Street 213, 50-556 Wroclaw, Poland; monika.sepek@umw.edu.pl (M.S.); krzysztof.kaliszewski@umw.edu.pl (K.K.); jerzy.rudnicki@umw.edu.pl (J.R.); 2Department of Dosage Form Technology, Wroclaw Medical University, Borowska Street 211 A, 50-556 Wroclaw, Poland; dominik.marciniak@umw.edu.pl; 3Infermedica Sp. z o.o., 50-062 Wroclaw, Poland; mateusz.glod@me.com

**Keywords:** complications, hypocalcemia, hypoparathyroidism, risk factors, thyroid surgery

## Abstract

Introduction: Iatrogenic hypoparathyroidism following thyroidectomy is one of the most common complications significantly reducing patients’ quality of life. Objectives: This study aimed to analyze the risk factors for calcium-phosphate disorders following thyroidectomy. Patients and methods: The study group consisted of 211 patients who underwent thyroidectomy for different conditions in 2018–2020. Demographic, clinical and surgical risk factors were analyzed against hypoparathyroidism and hypocalcemia. Results: Hypoparathyroidism occurred in 15.63% of patients, and hypocalcemia occurred in 45% of those operated on. There was statistical significance between hypoparathyroidism and the extent of thyroid surgery: thyroidectomy vs. lobectomy (*p* = 0.02, OR = 4.5) and surgeon experience (*p* = 0.016, OR = 6.9). Low preoperative PTH levels were associated with a higher incidence of hypoparathyroidism (*p* = 0.055, OR = 0.9). There was a statistically significant correlation between the occurrence of hypocalcemia and preoperative vitamin D deficiency (*p* = 0.04, OR = 3.5). Low calcium levels before surgery were associated with a higher incidence of hypocalcemia (*p* = 0.051, OR = 0.5). Meta-analyses of selected risk factors confirmed that the most significant factor in the incidence of hypocalcemia was a decrease in PTH levels (*p* < 0.001). Conclusions: The extent of thyroid surgery and the experience of the surgeon are the most significant risk factors for hypoparathyroidism. Hypocalcemia is much more common than hypoparathyroidism. Among the risk factors for hypocalcemia, in addition to the decrease in parathormone levels due to iatrogenic parathyroid damage, we should mention vitamin D deficiency in the preoperative period.

## 1. Introduction

Thyroid surgery is one of the most commonly performed procedures in general surgery. Total excision of the thyroid gland is a safe procedure, although not without complications, such as phonation disorders, hemorrhage requiring reoperation, and calcium-phosphate disorders, which significantly reduce the quality of life after thyroidectomy [1].

According to the 2019 Population Health Report for Poland, there has been a steady increase in the incidence of thyroid disease over the past five years; it affects 16% of women and 3% of men [2]. In addition, thyroid cancer has been on the rise over the past two decades; it is now the 6th most common cancer in women and the 21st in men. The incidence of thyroid cancer in Poland in 2020 was 10.6 per 100,000 population [3]. Undoubtedly, the observed increase in the incidence of thyroid disease is related to the greater availability of thyroid ultrasound examinations and fine-needle aspiration biopsy (FNA).

Surgical treatment of thyroid diseases is an important part of the treatment of thyroid diseases, apart from conservative treatment and radioactive iodine therapy; in some patients, it is the treatment of choice. In Poland, over 30,000 thyroid surgeries are performed annually, 12–15% of which are due to thyroid cancer; it is one of the most common surgical procedures performed in general surgery [4]. Over the centuries, the technique of thyroid surgery has evolved becoming more and more safe; among other things, the risk of thyrotoxic crisis has been practically completely eliminated. Over the past two decades, new technologies have been implemented, such as monitoring of the recurrent laryngeal nerves or the use of techniques enabling intraoperative identification of the parathyroid glands. Undoubtedly, it improved the quality of surgical treatment for thyroid diseases; nevertheless, complications after thyroid surgeries still occur [1].

Postoperative hypothyroidism, as a consequence of treatment, is a more common complication than hypoparathyroidism, which occurs less frequently than hypocalcemia [5,6,7]. A 2014 meta-analysis by Edafe et al. showed that transient hypoparathyroidism affected 19–38% of patients, while 0–3% of patients were permanently affected [8].

Hypoparathyroidism is not equivalent to the occurrence of hypocalcemia in the immediate period after thyroidectomy.

The 2017 American Thyroid Society guidelines provide a definition of hypoparathyroidism: biochemical and clinical, hypocalcemia and parathyroid failure [5]. A decrease in PTH (parathormone) levels below 15 pg/mL with concomitant hypocalcemia defines the group of patients with biochemical hypoparathyroidism, while patients with a decrease in calcium levels below 8.8 mg/dL should be classified in the group of patients with hypocalcemia. This condition can occur independently of hypoparathyroidism. Risk factors for post-thyroidectomy hypoparathyroidism include bilateral thyroid operations, autoimmune thyroid disease, central neck dissection, substernal goiter, surgeon inexperience, and malabsorptive conditions. Medical and surgical strategies to minimize perioperative hypoparathyroidism include optimizing vitamin D levels, preserving parathyroid blood supply, and autotransplanting ischemic parathyroid glands [5].

Constant monitoring of complications after thyroid surgeries and understanding the risk factors for their occurrence, along with multifactorial analysis, is the key to improving the quality of surgical treatment of thyroid diseases in a surgical department.

The purpose of this study was to analyze risk factors for calcium-phosphate diseases following thyroidectomy (hypoparathyroidism and hypocalcemia) separately in the postoperative period.

## 2. Patients and Methods

### 2.1. Study Design

Parathormone total calcium (Ca), phosphorus (P) and vitamin D_3_ (25-hydroxyvita min D) were determined in each patient before thyroidectomy and on the second postoperative day; clinical symptoms resulting from hypocalcemia were monitored. Demographic, clinical and surgical treatment data were collected in an Excel database.

Two endpoints of the study were identified: postoperative hypoparathyroidism and hypocalcemia, which were correlated with risk factors for complications.

Hypoparathyroidism was defined as a drop in PTH levels below 15 pg/mL, while hypocalcemia was diagnosed when Ca levels fell below 8.8 mg/dL according to the 2017 American Thyroid Society guidelines [5]. Patients who developed hypoparathyroidism following thyroidectomy were monitored for 6 months. Patients who successfully discontinued calcium supplements before a six-month period without recurrence of tetany symptoms were diagnosed with transient hypoparathyroidism. Permanent hypoparathyroidism was diagnosed if calcium supplementation had to be maintained beyond 6 months.

The following risk factors (demographics, dependent on thyroid disease, related to surgical treatment and “human factor”) were considered in this study.

(1)Demographics: age, gender, BMI;(2)Dependent on thyroid disease: clinical diagnosis, coexistence of autoimmune diseases (positive anti-TPO, anti-TG or TRAB antibodies), type of focal thyroid lesions: single vs. multiple vs. parenchymal goiter, presence of retrosternal goiter, tracheal displacement or narrowing;(3)Related to surgical treatment: type of surgery: primary vs. secondary, extent of thyroid surgery: total vs. partial surgery;(4)Human factor: operator experience (up to 50 thyroidectomy/year, more than 50 thyroidectomy/year);(5)Vitamin D deficiency (diagnosed when the concentration 25-hydroxyvitamin D in the blood was lower than *n* < 30 nmol/L) was considered a risk factor for complications, and changes in Ca, P and PTH levels before vs. after surgical treatment were analyzed.

### 2.2. Inclusion Criteria

The medical records of 211 patients who underwent thyroidectomy for different disorders in 2018–2020 at the Department of General, Minimally Invasive and Endocrine Surgery of the Medical University of Wroclaw were analyzed retrospectively.

### 2.3. Exclusion Criteria

Those with hypoparathyroidism or hypocalcemia diagnosed and treated before thyroidectomy were excluded.

On 28 February 2019, approval from the Bioethics Committee (KB—156/2019) for the study was obtained.

### 2.4. Statistical Analysis

Statistical analyses were carried out using the computer program STATISTICA PL^®^ version 13.3 with the add-on—Set Plus version 3.0.

Pearson’s non-parametric chi^2^ test was used to assess the statistical significance of correlations between variables on nominal scales; two types of statistical analyses were used to assess the statistical significance of correlations between dichotomous and quotient variables: parametric Student’s *t*-test for independent samples and univariate logistic regression. The general linear model (GLM) of Gauss–Markov was used to estimate the parameters of the logistic function.

A parametric one-way analysis of variance (ANOVA) was used to statistically evaluate the correlation between nominal variables, whose number of categories was greater than 2, and quantitative variables. The results of Student’s *t*-tests and parametric ANOVA were illustrated with standard box-and-whisker plots (mean SR; mean SR ± standard error; mean SR ± 1.96 standard error).

Multivariate analyses based on dimension reduction were performed using the procedures of decomposing the matrix of results according to singular values. Principal component analysis (PCA) and correspondence analysis were used to preliminarily identify the overall relationships between the statistically evaluated variables. The results of the statistical analyses were finally subjected to meta-analysis. In each case, the variable effects meta-analysis model was used. The performed meta-analysis was supplemented by a sensitivity analysis, which made it possible to assess the impact of each of the analyzed variables separately on the statistical significance of the constructed meta-analysis models. A significance level of *α* = 0.05 was assumed in all statistical analyses performed.

In the case of the statistical significance (*p* < 0.05) of the test chi^2^ evaluating the relationship between the dichotomous variables listed in Tables 5 and 7, to visualize the obtained results, OR odds ratios were additionally determined. The values were calculated using the standard method based on the constructed 2 × 2 two-way tables: Tab = {[a,b], [c,d]} V (a/c)/(b/d).

The statistical significance of correlations between dichotomous variables and continuous variables presented in Tables 4 and 6 were visualized by determining odds ratios with 95% significance intervals ± 95% CI. In this case, the values were estimated using one-way logistic regression. Those for which the range ± 95% CI did not include 1 were considered statistically significant.

## 3. Results

The demographic data and clinical characteristics of the patients are shown in Table 1.

Of the 211 patients who underwent thyroidectomy for different conditions, hypoparathyroidism was observed in 27 patients, with 12.79 cases per 100 patients operated on.

Hypoparathyroidism occurred in 25 (15.63%) of 160 patients who underwent total or near-total thyroid resections and in 2 (3.92%) of 51 patients who underwent lobectomy. Univariate analysis with Pearson’s non-parametric chi^2^ test showed a statistically significant correlation between the incidence of hypoparathyroidism and the extent of thyroid surgery (*p* = 0.029, OR = 4.5). In the group of patients whose surgery involved both lobes of the thyroid gland, the risk of hypoparathyroidism was more than four times higher than in the patients who underwent lobectomy.

Further statistical analysis regarding the influence of risk factors on the incidence of hypoparathyroidism and hypocalcemia was carried out in a group of 160 patients whose surgery involved only both lobes of the thyroid gland. This made it possible to standardize the study group by eliminating the influence of a strong, statistically significant factor—the extent of surgery—on the thyroid gland.

The mean PTH level before thyroid surgery was 62.12 (±25.25) pg/mL, and after surgery, it was 42.04 (±26.40) pg/mL (*p* < 0.001). The Ca level was 9.63 (±0.71) mg/dL before surgery and 8.76 (±0.84) mg/dL after surgery (*p* < 0.001), and the phosphorus level was 3.59 (±0.68) mg/dL before surgery and 3.71 (±0.82) mg/dL after thyroid surgery (*p* = 0.13).

Table 2 shows the incidence of hypoparathyroidism and hypocalcemia among 160 patients who underwent total thyroid gland resection, taking into account the characteristics of these groups. Table 3 shows the risk factors for hypoparathyroidism and hypocalcemia, along with the percentage of complications in the risk groups and the percentage distribution of complications by risk factor.

### 3.1. Risk Factors vs. Postoperative Hypoparathyroidism

Table 4 shows the effect of risk factors expressed in quotient scales on the incidence of hypoparathyroidism with univariate logistic regression, while Table 5 shows the effect of risk factors expressed in nominal scales on hypoparathyroidism with univariate analysis and Pearson’s non-parametric chi^2^ test.

Univariate analysis with Pearson’s non-parametric chi^2^ test showed a statistically significant correlation between the occurrence of hypoparathyroidism and surgeon experience (*p* = 0.017, OR = 6.99). The odds ratio OR = 6.99 indicates that in the group of patients operated on by an experienced surgeon, the risk of hypoparathyroidism was almost seven times lower than when the surgery was performed by a surgeon with less experience. The percentage of hypoparathyroidism in the group of patients operated on by experienced surgeons was 11.67%, and it was 27.5% in the group with less experienced surgeons.

Univariate analysis by Pearson’s non-parametric chi^2^ test showed no statistical significance (*p* > 0.05) between the incidence of hypoparathyroidism and risk factors, such as gender, type of thyroid surgery clinical diagnosis, autoimmune disease comorbidity, nature of focal lesions, tracheal displacement/narrowing, retrosternal goiter, extent of thyroid surgery and vitamin D_3_ levels. Selected conclusions emerging from the analysis of the percentage of complications cannot be overlooked. In the group of patients undergoing secondary surgery, a drop in PTH levels below 15 pg/mL affected 40% of those operated on; for primary surgery, it was 14.84%. The most common decrease in PTH levels affected patients operated on for thyroid cancer, at 18.18%. Of the patients who developed hypoparathyroidism, 70% were overweight or obese. More than 60% of patients with hypoparathyroidism had vitamin D deficiency before surgery; in the group with vitamin D deficiency, the incidence of hypoparathyroidism was 19%, compared to only 11% in patients with normal vitamin D_3_ levels.

A logistic regression model for quotient variables showed, at the limit of statistical significance, a relationship between the incidence of hypoparathyroidism and preoperative PTH levels. Low PTH concentration predisposed to a higher incidence of hypoparathyroidism (*p* = 0.055, OR = 0.98, OR − 95% CI-0.96, OR + 95% CI-1.00).

Studies on the influence of selected risk factors on hypoparathyroidism were summarized in meta-analyses performed, which confirmed the key influence of surgeon experience on the occurrence of the complication in question.

The interrelationships between risk factors and hypoparathyroidism and hypocalcemia were analyzed using multivariate “data mining” techniques based on dimension reduction of principal component analysis (PCA) and correspondence analysis, reducing the number of dimensions to two (PC1 and PC2). Analysis of the scatter plot of eigenvector loadings showed that hypoparathyroidism did not coincide with the location of hypocalcemia (Figure 1). This prompted separate analyses: risk factors vs. hypoparathyroidism and risk factors vs. hypocalcemia.

### 3.2. Risk Factors vs. Hypocalcemia

Table 6 shows the effect of risk factors expressed in quotient scales on hypocalcemia (univariate logistic regression), and Table 7 shows the effect of risk factors expressed in nominal scales on hypocalcemia using univariate analysis with Pearson’s non-parametric chi^2^ test.

Univariate analysis using Pearson’s non-parametric chi^2^ test showed a statistically significant relationship between the occurrence of hypocalcemia and vitamin D deficiency (*p* = 0.049, OR = 3.55). Hypocalcemia was observed three times more often in the group of patients with vitamin D deficiency than among patients with normal levels. A statistically significant correlation between hypocalcemia and intraoperative parathyroid injury was confirmed (*p* < 0.001).

Univariate analysis using Pearson’s non-parametric chi^2^ test showed no statistical significance (*p* > 0.05) between the incidence of hypocalcemia and risk factors, such as gender, clinical diagnosis, type of thyroid surgery, autoimmune disease comorbidity, tracheal displacement/narrowing, retrosternal goiter, nature of focal thyroid lesions and extent of thyroid surgery.

A logistic regression model for quotient variables showed a relationship between the incidence of postoperative hypocalcemia and total calcium concentration before thyroid surgery at the limit of statistical significance. A low preoperative Ca concentration had an affinity for a higher incidence of postoperative hypocalcemia (*p* = 0.052, OR = 0.53, OR − 95% CI-0.28, OR + 95% CI-1.00).

Studies on the effect of selected risk factors on hypocalcemia were summarized in meta-analyses, which confirmed that the most significant factor in the occurrence of postoperative hypocalcemia was a decrease in PTH levels due to intraoperative parathyroid injury.

## 4. Discussion

Disorders in calcium and phosphate metabolism following thyroidectomy significantly affect patients’ quality of life. While hypoparathyroidism almost always results in a decrease in calcium concentration, postoperative hypocalcemia is more common [5,9,10,11,12,13,14,15,16,17,18,19,20]. There are different thresholds for Ca and PTH concentrations, as well as different timing of calcium, phosphorus and PTH determinations after thyroid surgery for the diagnosis and monitoring of these complications [5]. The authors of this publication based their analysis of the impact of risk factors on hypoparathyroidism and hypocalcemia on the 2017 ATA definitions [5].

The clinical material showed that after thyroid surgery, patients had a statistically significant decrease in both PTH and Ca levels (*p* < 0.001), accompanied by an increase in *p* levels, albeit not at the level of statistical significance (*p* = 0.135).

Hypoparathyroidism occurred in 15.63% of patients and was accompanied by hypocalcemia in 92%. Hypocalcemia was observed in almost half of the patients (45%), but only in 31% of the patients was the cause PTH deficiency. Clinical symptoms requiring treatment were observed in 76% of patients with a decline in biochemical PTH, while in the group of patients with hypocalcemia, they affected 29.17%. Although the clinical manifestations of both of these complications are the same, as they result from a deficiency in calcium levels, their etiology is different, and their risk factors may differ. Hence, it seems expedient to separately analyze the risk factors for hypoparathyroidism and hypocalcemia.

Transient hypoparathyroidism ranges from 7% to 51% [5,6,7,10,21,22,23], of which permanent affects 0–5% of those operated on [5,8,24]. Among the risk factors for hypoparathyroidism are bilateral thyroid gland surgery, concomitant autoimmune diseases, thyroid surgery with lymphadenectomy, little surgical experience, secondary surgery, and retrosternal goiter [5,9,10,11,16].

The first stage of the study considered the effect of the extent of surgery on hypoparathyroidism. In the group of patients whose surgery involved both lobes of the thyroid gland, the risk of hypoparathyroidism was more than four times higher than in those who underwent lobectomy. This result was expected, since by performing a lobectomy, a minimum of two parathyroid glands remain intact, and this is sufficient for proper maintenance of PTH levels. The results obtained are in line with many publications in which the extent of surgery is the primary risk factor for hypoparathyroidism [5,9,10,11,16]. It seems advisable to limit the extent of thyroid surgery to one lobe with the parathyroid gland whenever possible in the absence of significant changes in the second lobe of the thyroid gland, especially in the case of thyroid microcarcinoma, which is in line with the recent trend in the treatment of thyroid cancer [5,25,26,27].

The second factor significantly affecting the occurrence of hypoparathyroidism was surgeon experience. Univariate analysis with Pearson’s non-parametric chi^2^ test showed a statistically significant correlation between the occurrence of hypoparathyroidism and surgeon experience (*p* = 0.016, OR = 6.99). In the group of patients operated on by an experienced surgeon, the risk of hypoparathyroidism was almost seven times lower than when the operation was performed by a surgeon with less experience. The percentage of hypoparathyroidism operated on by experienced surgeons was 11.67%; in the group where the operation was performed by less experienced surgeons, it was 27.5%. In the clinic from which the study material came, surgical experience is due to the large number of thyroid surgeries performed; magnifying glasses were not used, nor were modern fluorescent techniques used to facilitate parathyroid identification [28,29,30,31,32]. Such an experience would be beneficial for parathyroid prevention, as evidenced in publications [28,29,30,31,32]. Pata et al. [33] noted that the use of magnifying glasses reduced the risk of hypoparathyroidism from 3.8% to 7.8% (*p* < 0.001). The authors use the technique of extracapsular preparation of the parathyroid gland by sparing the surrounding adipose tissue, with special attention to the delicate vascularization of the parathyroid glands. If the parathyroid glands could not be safely dissected, the technique of excising the parathyroid gland “on a patch” formed from a fragment of the thyroid gland was used. Surgeon experience as an important risk factor for hypoparathyroidism has been discussed in many publications [10,11,16]. Cocchiara et al. [34] showed that surgical experience is the most significant risk factor for a decrease in PTH levels. Additionally, Anagnostisa et al. [35] pointed out a significant difference in the number of complications depending on the surgeon’s experience. The authors showed that the rate of hypoparathyroidism was statistically significantly higher in the group of surgeons with little experience (*p* < 0.001). Moreover, they showed that the consequence of thyroid surgery performed by surgeons with less experience was an increase in the cost of surgery from 23% to 45% due to the need for supplementation of calcium preparations and vitamin D_3_. Additionally, Paduraru, Sua, and Falch et al. showed that experience and skillful techniques in isolating the parathyroid glands from the thyroid gland capsule affect the incidence of this complication [12,13,36]. In contrast, Asarii et al. did not confirm that the surgeon’s experience had a significant effect on the transient and permanent occurrence of this complication [9]. The risk factor of surgeon experience is very difficult to analyze objectively; more experienced surgeons usually perform thyroid surgery with a higher risk of complications.

Other risk factors, such as age, gender of patients, clinical diagnosis, malignant neoplastic conditions, and secondary surgery on the thyroid gland, were not a significant factor in the occurrence of hypoparathyroidism (*p* > 0.005). Most likely due to the small size of the study group, the above factors were not shown to be significant against hypoparathyroidism. These results are quite discrepant with extensive studies by other authors [5,13,14,17,20], who noted an increase in the rate of this complication in the group of patients with total thyroid resection with lymphadenectomy, thyroid cancer, Graves-Basedov disease, or retrosternal and recurrent goiter.

Summarizing the discussion of hypoparathyroidism, some limitations should be noted: there was no analysis of risk factors for permanent hypoparathyroidism due to the small size of the study group. Hence, the results obtained should be referred to as early complications in the immediate period following thyroidectomy. The extent of thyroid surgery and the surgeon’s experience have been shown to have a significant impact on the occurrence of this complication. Undoubtedly, very different surgical experience—both in the number of operations performed and in the years of practice—is another factor that may have affected the interpretation of the examination. The very small number of patients with secondary surgery is another limitation of this study. Secondary operations on the thyroid gland are undoubtedly a significant risk factor for complications, although demonstrating statistical significance would require a multi-center study on a much larger group of patients.

Hypocalcemia is a much broader concept than postoperative hypoparathyroidism [18,19,20,37,38]. Transient hypocalcemia occurs from 1.6% of those operated on to 53% [39,40], permanent one affects an average of 1% of patients (0–3%) [8,41].

The etiology of hypocalcemia after thyroid surgery is multifactorial [42], and the most common factors that increase its risk include hypoparathyroidism, total thyroid resection, hypomagnesemia, preoperative vitamin D deficiency, female gender, thyroid cancers, thyroiditis, multinodular goiter, incidental parathyroid gland resection, surgery with neck lymph node resection, and surgeon experience [12,13,14,15,17].

In the first step, we analyzed the effect of the extent of thyroid surgery on the occurrence of hypocalcemia. Univariate analysis with Pearson’s non-parametric chi^2^ test showed no statistically significant correlation between the occurrence of hypocalcemia and the extent of surgery (*p* = 0.307), although it was almost three times more common in the group of patients with total thyroid resection than after lobectomy. While this factor had a significant impact on hypoparathyroidism, the role of the extent of surgery was much less in the case of hypocalcemia, reinforcing our belief that the two complications should not be treated equally.

The study showed that two risk factors had a statistically significant effect on the rate of hypocalcemia. These were hypoparathyroidism and preoperatively diagnosed vitamin D deficiency.

Researchers agree that hypoparathyroidism is a major cause of hypocalcemia [12,13,14,15,17,19], although it is not the only one. Univariate analysis with Pearson’s nonparametric chi^2^ test showed that hypoparathyroidism was the most significant risk factor for calcium decline after thyroid surgery (*p* = 0.001, OR = 26). One-dimensional logistic regression showed that a decrease in PTH levels and an increase in phosphorus levels correlated at the level of statistical significance (*p* = 0.006; *p* ≤ 0.001) with hypocalcemia; hence, the determination of these parameters should be considered as sensitive indicators of this complication. Both the effect of bilateral surgical extent on hypoparathyroidism and the close relationship between iatrogenic parathyroid damage and decreased calcium levels were not surprising, as daily clinical practice confirms these relationships.

An interesting issue is the effect of vitamin D deficiency on hypocalcemia [18,19,20,43]. Vitamin D deficiency is a common worldwide problem, as pointed out by Hollick et al. [44,45]. The overlap of vitamin D_3_ deficiency along with iatrogenic parathyroid damage, may exacerbate hypocalcemia in the postoperative period [18,19,20,45]. To date, the role of vitamin D_3_ in predicting hypoparathyroidism has not been clearly elucidated [18,19,20,46,47,48].

In this study, more than half of the patients (52.5%) had vitamin D deficiency. The prevalence of hypocalcemia in patients with vitamin D deficiency was 65.28%, which was higher than the rate of hypocalcemia in patients with normal levels (34.72%). Preoperative vitamin D deficiency was a significant factor in the occurrence of hypocalcemia (*p* = 0.048, OR = 3.55). Patients with vitamin D deficiency were more than three times more likely to develop hypocalcemia. Interestingly, the analysis showed that preoperative calcium levels, at the limit of statistical significance (*p* = 0.051), correlated with the occurrence of hypocalcemia, which is in line with the correlations arising from vitamin D deficiency and its effects on calcium metabolism. Almost identical results were obtained by Bove et al. [20], showing that vitamin D deficiency was a predictor of both biochemical (*p* = 0.012) and clinical hypocalcemia (*p* = 0.045), and the risk of its occurrence was 15 or 18 times higher than in patients with normal calcium levels, respectively. Additionally, a 2016 study by Daglar et al. [18], confirmed that both preoperative calcium and vitamin D levels were statistically significant factors in the occurrence of hypocalcemia (*p* < 0.05). Erbil et al. analyzed different thresholds for vitamin D deficiency and both severe and moderate vitamin D deficiency had a significant effect on the occurrence of hypocalcemia [49,50]. Kirby-Bott et al., on the other hand, not only pointed out that vitamin D deficiency was associated with hypocalcemia but also showed that levels above 50 ng/mL were a factor in preventing hypocalcemia [47]. In meta-analyses, Edafe et al. in 2017 [8], Qin et al. in 2020 [14] and Chen et al. in 2021 [17] indicated that vitamin D deficiency is a significant factor in the occurrence of hypocalcemia after thyroid surgery.

In contrast, the publication by Singh et al. [19] showed that vitamin D_3_ deficiency did not affect the incidence of hypocalcemia (*p* = 0.352). Additionally, studies by Cherian [51] Griffin [46], Lin [52], and Chia [53] did not confirm the association of the occurrence of hypocalcemia with preoperative vitamin D deficiency. Such discrepant results may be due to the different cutoff points the authors adopted when assessing vitamin D_3_ deficiency; moreover, the influence of surgeon’s experience also could not be eliminated.

To summarize the results of our own research and that of other authors [20,47,49,50], we can confirm the important role of adequate vitamin D_3_ levels (>30 ng/mL) in maintaining calcium homeostasis, especially in the period before surgical treatment of thyroid gland disorders. It seems reasonable to compensate for vitamin D deficiency before surgical treatment [14,17,18,20]. However, not all authors [19] agree that compensating for vitamin D_3_ levels can protect patients from this complication. The limitation of this part of the study is its retrospective nature; both the lack of information on whether patients periodically or constantly supplemented vitamin D prior to surgery and time occurrence of its administration.

The results of the study did not confirm the effect of gender, age, BMI, clinical diagnosis, or the presence of retrosternal goiter or autoimmune disease on the rate of postoperative hypocalcemia (*p* > 0.05). The work of Singh et al. [19] also showed no association of gender, age, BMI or race with increased rates of hypocalcemia. Bove et al. also found no association between gender, age, BMI or hyperthyroidism and a higher incidence of calcium drops after thyroidectomy [20]. In contrast, in large meta-analyses, Chen [17], Qin [14], and Edafe [8] showed that younger patient age and female gender correlated statistically significantly more frequently with the occurrence of hypocalcemia [14,17]. The most comprehensive meta-analysis by Chen [17] from 2021 showed that hypocalcemia occurred almost three times more often in patients with magnesium deficiency, twice as often in patients treated for thyroid cancer, with concomitant thyroiditis, retrosternal goiter and in operations with concomitant lymphadenectomy. Although the results did not show the statistical significance of the demographic or clinical risk factors discussed above, the prevalence of hypocalcemia in younger patients was 76.39% vs. 23.61% in patients older than 65 years; moreover, it was more common in women (84.72%) than in men (15.28%), which is consistent with the trend described in the meta-analyses above.

Understanding the risk factors for hypoparathyroidism and hypocalcemia is important in preventing these complications. It affects the proper, prudent selection of the extent of surgery, as well as qualifying patients early enough for surgical treatment. It also affects the care of surgical techniques and indicates those thyroidectomies that should be performed by surgeons with extensive experience. Undoubtedly, the analysis of complications in a given center and the analysis of risk factors improves the quality of surgical treatment in the surgery unit.

## 5. Conclusions

Hypocalcemia is a broader concept than hypoparathyroidism following thyroidectomy. Both complications are closely related; the main risk factor for hypocalcemia is iatrogenic parathyroid injury, while for hypoparathyroidism, it is the extent of thyroid surgery. The demonstrated relationship between vitamin D deficiency and the incidence of hypocalcemia is significant, and it provides an opportunity to prevent complications by balancing vitamin D before surgery.

## Figures and Tables

**Figure 1 biomedicines-11-02299-f001:**
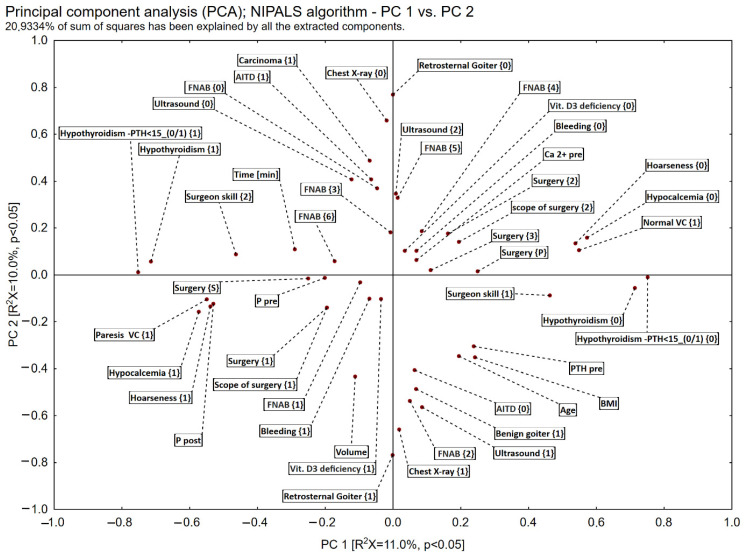
Postoperative hypoparathyroidism and hypocalcemia—principal component analysis—PCA, The *X*-axis represents dependency between the risk factors and postoperative hypoparathyroidism (PC 1). The *Y*-axis represents the dependency between the risk factors and postoperative hypocalcemia (PC 2).

**Table 1 biomedicines-11-02299-t001:** Clinical characteristics of patients.

Clinical Characteristics of Patients	*n* (%)
Number of patients	211 (100%)
Number of RLNs at risk of damage, *n* (%)	371 (100%)
Age, mean ± standard deviation, median years Minimum/maximum age, years	51.7 ± 14.54; 52 16/82
Gender (Female: Male)	165:46 (3.6:1)
BMI, mean ± standard deviation, median kg/m^2^ BMI minimum/maximum	28.07 ± 5.7; 27.5 16.26/44.96
Thyroid volume (V), mean ± standard deviation, median mL V minimum/maximum, mL	44.35 ± 57.3; 29.6 3.6/650
Morphological changes of the thyroid gland, *n* (%)	
– Single focal change	39 (18.5%)
– Focal lesions bilateral	157 (74.5%)
– Parenchymal goiter	15 (7%)
Retrosternal goiter, *n* (%)	67 (31.8%)
Tracheal displacement/overload, *n* (%)	51 (24.2%)
Diagnosis, *n* (%)	
Multinodular goiter	150 (71.1%)
Toxic multinodular goiter	14 (6.65%)
Graves-Basedow disease *	18 (8.5%)
Thyroid cancer	29 (13.75%)
Primary operation, *n* (%)	201 (95.3%)
Secondary surgery, *n* (%)	10 (4.7%)
Range of operations, *n* (%)	
Total excision of the thyroid gland	156 (73.93%)
Subtotal excision of both lobes of the thyroid gland	2 (0.95%)
Dunhill operation	2 (0.95%)
Hemithyroidectomy	51 (24.17%)
Operation time, mean ± standard deviation, median min Shortest/longest time	97.87 ± 31.13, 95 30/185
Surgeon’s experience, *n* (%)	
more than 50 thyroidectomy/year	156 (73.93%)
up to 50 thyroidectomy/year	55 (26.07%)

Abbreviations: *n*—group size; RLN—recurrent laryngeal nerve. * For confirmation of autoimmune thyroid disorders (AITD): anti-TPO (*n* < 35,0 IU/mL), anti-Tg: (*n* < 4.11 IU/mL), TRAB (*n*< 1.8 IU/mL).

**Table 2 biomedicines-11-02299-t002:** Characteristics of patients with hypoparathyroidism and hypocalcemia following thyroidectomy.

Number of Patients with Bilateral Thyroid Resection, *n* (%)	160	(100%)	100%
Postoperative hypoparathyroidism in the immediate post-thyroid surgery period (PTH < 15 pg/mL), *n* (%)	25		15.63%
– With clinical signs of tetany	19	76%	11.88%
– With associated hypocalcemia (Ca < 8.8 mg/dL)	23	92%	14.38%
– With associated hyperphosphatemia (P > 4.5 mg/dL)	9	36%	5.63%
– Vitamin D deficiency (Vit. D_3_ < 30 ng/mL)	16	64%	10%
Postoperative permanent hypoparathyroidism	3	(100%)	1.875%
Postoperative hypocalcemia (Ca < 8.8 mg/dL), *n* (%)	72	100%	45%
– With clinical signs of tetany	21	29.17%	13.13%
– With a drop in PTH < 15 pg/mL	23	31.94%	14.38%
– With associated hyperphosphatemia (P > 4.5 mg/dL)	19	26.39%	11.88%
– Vitamin D deficiency (Vit. D_3_ < 30 ng/mL)	43	55.72%	26.88%
Postoperative hypocalcemia persistent	5	100%	3.125%

Abbreviations: Ca, total calcium; P, phosphorus; PTH, parathormone; vit. D_3_, vitamin D_3_.

**Table 3 biomedicines-11-02299-t003:** Risk factors for hypoparathyroidism and hypocalcemia.

Risk Factors for Complications		Number of Patients *n* = 160 (100%)	Hypoparathyroidism PTH < 15 pg/mL		Hypocalcemia Ca < 8.8 mg/dL	
			Number of Patients at Risk *n* (100%)	Number of Patients *n* = 25 (100%)	Number of Patients at Risk *n* (100%)	Number of Patients *n* = 72 (100%)
Age	<65 years	121 (75.63%)	17 (14.05%)	68%	55 (45.45%)	76.39%
	≥65 years	39 (24.37%)	8 (20.51%)	32%	17 (43.59%)	23.61%
Gender	Women	129 (80.63%)	20 (15.5%)	80%	61 (47.29%)	84.72%
	Men	31 (19.37%)	5 (16.13%)	20%	11 (35.48%)	15.28%
BMI (kg/m^2^)	Underweight (≤18.5)	2 (1.25%)	1 (50%)	4%	2 (100%)	2.78%
	Normal body weight (18.5–24.9)	50 (31.25%)	10 (20%)	40%	26 (52%)	36.11%
	Overweight (25–29.9)	62 (38.75%)	9 (14.52%)	36%	28 (45.16%)	38.89%
	Obesity (>30)	46 (28.75%)	5 (10.87%)	20%	16 (34.78%)	22.22%
Clinical diagnosis	Nodular goiter	106 (66.25%)	18 (16.98%)	72%	47 (44.34%)	65.28%
	Toxic nodular goiter	14 (8.75%)	1 (7.14%)	4%	9 (64.29%)	12.5%
	Graves- Basedow disease	18 (11.25%)	2 (11.11%)	8%	9 (50.00%)	12.5%
	Thyroid cancer	22 (13.75%)	4 (18.18%)	16%	7 (31.82%)	9.72%
Focal changes in the thyroid gland	Single	16 (10%)	3 (18.75%)	12%	7 (43.75%)	9.72%
	Plural	129 (80.63%)	21 (16.28%)	84%	58 (44.96%)	80.56%
	Parenchymal goiter	15 (9.37%)	1 (6.67%)	4%	7 (46.67%)	9.72%
Total volume	≤25 mL	47 (29.3%)	6 (12.77%)	24%	24 (51.06%)	33.33%
	25–50 mL	73 (45.63%)	13 (17.81%)	52%	25 (34.25%)	34.73%
	>50 mL	40 (25.00%)	6 (15.0%)	24%	23 (57.5%)	31.94%
Displaced/constricted trachea		42 (26.25%)	5 (11.90%)	20%	22 (52.38%)	30.55%
Retrosternal goiter		59 (36.87%)	9 (15.25%)	36%	30 (50.85%)	41.67%
Autoimmune disease		25 (15.62%)	3 (12.0%)	12%	14 (56.0%)	19.4%
Operation	Primary	155 (96.88%)	23 (14.84%)	92%	69 (44.52%)	95.83%
	Secondary	5 (3.12%)	2 (40.0%)	8%	3 (60.0%)	4.17%
Scope of thyroid surgery	Total	156 (97.5%)	25 (16.03%)	100%	71 (45.51%)	98.61%
	Partial	4 (2.5%)	0 (0%)	0%	1 (25.0%)	1.39%
Experienced surgeon	≤50 operation/year	40 (25.0%)	11 (27.5%)	44%	22 (55.0%)	30.55%
	>50 operation/year	120 (75.0%)	14 (11.67%)	56%	50 (41.67%)	69.55%
Vitamin D_3_ levels before surgery, ng/mL	Deficiency [≤30].	84 (52.5%)	16 (19.05%)	64%	47 (55.95%)	65.28%
	Normal level [>30].	76 (47.5%)	9 (11.84%)	36%	25 (32.89%)	34.72%

Abbreviations: BMI, body mass index; Ca, total calcium; *n*—group size; PTH, parathormone; vit. D_3_, vitamin D_3_.

**Table 4 biomedicines-11-02299-t004:** Risk factors for hypoparathyroidism following thyroidectomy—univariate logistic regression.

Risk Factors for Complications		Hypoparathyroidism PTH < 15 pg/mL			
		*p*	Odds Quotient OR	OR − 95% CI	OR + 95% CI
Age of patients	word free	0.100	0.27	0.05	1.28
	regression coefficient	0.617	0.99	0.96	1.02
BMI	word free	0.685	0.62	0.06	5.95
	regression coefficient	0.286	0.95	0.88	1.03
Thyroid volume	word free	0.000	0.18	0.10	0.31
	regression coefficient	0.986	1.00	0.99	1.00
Duration of operations	word free	0.008	0.11	0.02	0.57
	regression coefficient	0.549	1.00	0.99	1.01
PTH levels prior to surgery	word free	0.366	0.58	0.17	1.88
	regression coefficient	0.055	0.98	0.96	1.00
Calcium levels before surgery	word free	0.252	0.00	0.00	28.66
	regression coefficient	0.460	1.36	0.59	3.12
Phosphorus levels prior to surgery	word free	0.273	0.25	0.02	2.94
	regression coefficient	0.796	0.91	0.46	1.79
Vitamin D_3_ levels before surgery	word free	0.023	0.25	0.07	0.83
	regression coefficient	0.606	0.99	0.95	1.02

Abbreviations: BMI, body mass index: PTH, parathormone.

**Table 5 biomedicines-11-02299-t005:** Risk factors for hypoparathyroidism following thyroidectomy—univariate analysis by Pearson’s non-parametric chi^2^ test.

Risk Factors for Complications	Number of Patients *n* (100%)		Level of Significance *p*	Odds Ratio OR
	Number of Patients *n* (%) With Hypoparathyroidism: PTH < 15 pg/mL			
Gender	Women	Men		
	129 (100%)	31 (100%)		
	20 (15.5%)	5 (16.13%)	0.931	2.77
Operation	Primary	Secondary		
	155 (100%)	5 (100%)		
	23 (14.84%)	2 (40.00%)	0.127	24.16
Clinical diagnosis	Nodular goitre/ Toxic nodular goiter	Graves-Basedow’s disease/Thyroid cancer		
	106 (100%) 14 (100%)	18 (100%) 22 (100%)		
	18 (16.98%) 1 (7.14%)	2 (11.11%) 4 (18.18%)	0.729	-
Autoimmune disease	YES	NO		
	25 (100%)	135 (100%)		
	3 (12.0%)	22 (16.30%)	0.586	2.54
Type of focal lesions in thyroid ultrasound	Single tumor	Multiple nodules/ Parenchymal goiter		
	16 (100%)	129 (100%) 15 (100%)		
	3 (18.75%)	21 (16.28%) 1 (6.67%)	0.584	-
Displaced trachea/ Constricted (X-ray)	YES	NO		
	42 (100%)	118 (100%)		
	5 (11.90%)	20 (16.95%)	0.439	1.89
Retrosternal goiter	YES	NO		
	59 (100%)	101 (5.71%)		
	9 (15.25%)	16 (15.84%)	0.921	2.32
Experienced surgeon	>50 operation/year	≤50 operation/year		
	120 (100%)	40 (100%)		
	114 (11.67%)	11 (27.5%)	0.016	6.99
Total operation	YES	NO		
	156 (100%)	4 (100%)		
	25 (16.03%)	0 (0%)	0.383	-
Vitamin D deficiency	YES	NO		
	84 (100%)	76 (100%)		
	16 (19.05%)	9 (11.84%)	0.210	4.23

Abbreviations: *n*—group size.

**Table 6 biomedicines-11-02299-t006:** Risk factors for hypocalcemia after thyroid surgery—univariate logistic regression.

Risk Factors for Complications		Postoperative Hypocalcemia (Ca < 8.8 mg/dL)			
		*p*	Odds Quotient OR	OR − 95% CI	OR + 95% CI
Age of patients	word free	0.312	1.82	0.56	5.90
	regression coefficient	0.163	0.98	0.96	1.00
BMI	word free	0.278	2.45	0.48	12.46
	regression coefficient	0.177	0.96	0.90	1.01
Thyroid volume	word free	0.041	0.62	0.39	0.98
	regression coefficient	0.125	1.00	0.99	1.01
Duration of operations	word free	0.050	0.30	0.09	1.00
	regression coefficient	0.090	1.00	0.99	1.02
PTH levels prior to surgery	word free	0.771	1.13	0.49	2.60
	regression coefficient	0.412	0.99	0.98	1.00
Calcium levels before surgery	word free	0.061	3.21	0.76	1.35
	regression coefficient	0.051	0.53	0.28	1.00
Phosphorus levels prior to surgery	word free	0.237	0.34	0.05	2.02
	regression coefficient	0.330	1.27	0.78	2.07
Vitamin D_3_ levels before surgery	word free	0.298	1.60	0.65	3.94
	regression coefficient	0.117	0.97	0.95	1.00

Abbreviations: BMI, body mass index; Ca, total calcium; PTH, parathormone.

**Table 7 biomedicines-11-02299-t007:** Risk factors for hypocalcemia after thyroid surgery—univariate analysis by Pearson’s non-parametric chi^2^ test.

Risk Factors for Complications	Number of Patients *n* (100%)		Significance Level *p*	Odds Ratio OR
	Number of Patients *n* (%) With Hypocalcemia (Ca < 8.8 mg/dL)			
Gender	Women	Men		
	129 (100%)	31 (100%)		
	61 (47.29%)	11 (35.48%)	0.235	3.67
Operation	Primary	Secondary		
	155 (100%)	5 (100%)		
	69 (44.52%)	3 (60.00%)	0.493	11.50
Clinical diagnosis	Nodular goitre/ Toxic nodular goiter	Graves-Basedow disease Thyroid cancer		
	106 (100%) 14 (100%)	18 (100%) 22 (100%)		
	47 (44.34%) 9 (64.29%)	9 (50.00%) 7 (31.82%)	0.278	-
Autoimmune disease	YES	NO		
	25 (100%)	135 (100%)		
	14 (56.0%)	58 (42.96%)	0.228	3.99
Focal changes in the thyroid gland	Single tumor	Multiple nodules/ Parenchymal goitre		
	16 (100%)	129 (100%) 15 (100%)		
	7 (43.75%)	58 (44.96%) 7 (46.67%)	0.986	-
Displaced trachea (X-ray)	YES	NO		
	42 (100%)	118 (100%)		
	22 (52.38%)	50 (42.37%)	0.262	3.03
Retrosternal goiter	YES	NO		
	59 (100%)	101 (100%)		
	30 (50.85%)	42 (41.58%)	0.255	2.77
Experienced surgeon	>50 operation/year	≤50 operation/year		
	120 (100%)	40 (100%)		
	50 (41.67%)	22 (55.0%%)	0.142	3.51
Total operation	YES	NO		
	156 (100%)	4 (100%)		
	71 (45.51%)	1 (25%)	0.415	3.92
Vitamin D deficiency	YES	NO		
	84 (100%)	76 (100%)		
	44 (52.38%)	28 (36.84%)	0.048	3.55
Hypoparathyroidism	YES	NO		
	25 (100%)	135 (100%)		
	23 (92.00%)	49 (30.63%)	*p* < 0.0001	89.27

Abbreviations: Ca, total calcium; *n*—group size.

## Data Availability

All research data are available in main author.

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
