# Peer review of "Risk Factors for Calcium-Phosphate Disorders after Thyroid Surgery"

_biomedicines, 2023, doi:10.3390/biomedicines11082299_

Round 1

Reviewer 1 Report

Dear Authors,

The topic of calcium and mineral metabolism following thyroidectomy is of great interest regardless endocrine or surgical background of a certain practitioner. That is why I consider the topic very important. The paper brings knowledge to our current understanding of such an important matter with practical applications.

Here are my observations or suggestions:

1.    Abstract. First sentence. Please do no repeat “after” and “surgery”. I suggest “following” and “thyroidectomy”.

2.    Abstract. Please use “patients who underwent thyroidectomy for different conditions (or disorders) instead of “ patients operated on due to thyroid gland disorders”

3.    Abstract. Lines 20-21. Please do no repeat words such as “post-operative”, etc. which are obvious.

4.    Abstract. “Low PTH levels had..” Do you mean pre-operatory PTH?

5.    Abstract. “had an affinity”. Do you mean “was associated with..”?

6.    When you mentioned correlations, you need to provide the correlation coefficient, as well (not just o value)

7.    Abstract. What do you mean by meta-analysis since this is an original study?

8.    We usually use the term of “vitamin D deficiency”, not “vitamin D3”..

9.    Introduction. Please do no repeat “disorders” within the same paragraph (I suggest “condition” or “disease”).

10. Introduction. Line 51. Actually post-operatory hypothyroidism is more common than hypoparathyroidism that is less often found than hypocalcemia

11. Methods. Please start with study design, then inclusion and exclusion criteria followed by assessments and then the ethical concerns and approvals.

12. Table 1. The characteristics of the patients are obtained after applying inclusion/exclusion criteria in one study thus I suggest moving the table as first section of Results.

13. Table 1. If you provide the minimum and maximum values, then median is advised, too.

14. Table 1. “n(%)” should be placed on the top of the second column.

15. Table 1. What do you mean by “K:M” at gender?

16. Data collection. Do you mean “25-hydroxyvitamin D”?

17. Methods. Lines 89-100. I suggest to either use a table or to introduce the data within a usual sentence

18. Please provide the details of assessments concerning “anti-TPO, anti-TG or TRAB antibodies” and how did you define a positive status.

19. Please define vitamin D deficiency (in terms of cut offs).

20. Table 1 and 2. Please use left alignment for the first column

21. Legend for Table 3. What do you mean for “risk factors versus…”?

22. Results. Lines 230-233. A meta-analysis is based on multiple studies, not just one study. Please re-organize the information.

23. Conclusion -  I suggest to use something other than “inexperience”, maybe defying the number of procedures (since a surgeon that performs 49 thyroidectomies per year  is not inexperienced while the surgeon performing 51 such procedures is “experienced”

Thank you

Still some aspects need to be refined. Thank you

Author Response

Dear Reviewer.

Thank you for your insightful and valuable comments on the manuscript. Below are our answers. We tried our best to improve the manuscript.

Kind regards,

Beata Wojtczak and Monika Sępek with the authors.

Dear Authors,

The topic of calcium and mineral metabolism following thyroidectomy is of great interest regardless endocrine or surgical background of a certain practitioner. That is why I consider the topic very important. The paper brings knowledge to our current understanding of such an important matter with practical applications.

Here are my observations or suggestions:

  1. Abstract. First sentence. Please do no repeat “after” and “surgery”. I suggest “following” and “thyroidectomy”. Ok, I have changed in many places “ after surgery” for “following thyroidectomy”

  1. Abstract. Please use “patients who underwent thyroidectomy for different conditions (or disorders) instead of “ patients operated on due to thyroid gland disorders” Ok, I have replaced “patients operated on due to thyroid gland disorders” into “patients who underwent thyroidectomy for different conditions’’.

  1. Abstract. Lines 20-21. Please do no repeat words such as “post-operative”, etc. which are obvious. Of course, in many places in the manuscript I have crossed out “post-operative”.
  2. Abstract. “Low PTH levels had..” Do you mean pre-operatory PTH? Yes, I corrected it in the text: Low preoperative PTH
  3. Abstract. “had an affinity”. Do you mean “was associated with..”? Yes, now I used: “ was associated with”.
  4. When you mentioned correlations, you need to provide the correlation coefficient, as well (not just o value) Of course, I have added a correlation coefficient.
  5. Abstract. What do you mean by meta-analysis since this is an original study?

This is an original study and the purpose of this study was to analyze risk factors for calcium-phosphate disorders following thyroidectomy. But in the end of our work  meta-analysiss of selected risk factors was performed: The results of the performed statistical analyses were finally subjected to meta-analysis. In each case, the variable effects meta-analysis model was used. The performed meta-analysis was additionally supplemented by a sensitivity analysis, which made it possible to assess the impact of each of the analyzed variables separately on the statistical significance of the constructed meta-analysis models. A significance level of α = 0.05 was assumed in all the statistical analyses performed. This has been clarified in the text

  1. We usually use the term of “vitamin D deficiency”, not “vitamin D3”. Of course, I have changed vitamin D3” into “vitamin D deficiency” in manuscript in all places where it should be.
  2. Introduction. Please do no repeat “disorders” within the same paragraph (I suggest “condition” or “disease”). Of course, I checked and changed.
  3. Introduction. Line 51. Actually post-operatory hypothyroidism is more common than hypoparathyroidism that is less often found than hypocalcemia. Of course, I agree I added this sentence to the manuscript.
  4. Methods. Please start with study design, then inclusion and exclusion criteria followed by assessments and then the ethical concerns and approvals. Ok, I have changed according suggestions.
  5. Table 1. The characteristics of the patients are obtained after applying inclusion/exclusion criteria in one study thus I suggest moving the table as first section of Results. Ok I have moved the table 1 to the results.
  6. Table 1. If you provide the minimum and maximum values, then median is advised, too. Ok I have added the median into the table.
  7. Table 1. “n(%)” should be placed on the top of the second column. Ok, I did it.
  8. Table 1. What do you mean by “K:M” at gender? Sorry it wasn’t translate at all. Now is correct, it should be: Gender (Female: Male)
  9. Data collection. Do you mean “25-hydroxyvitamin D”? It was: 25-hydroxyvitamin D. I precised in manuscript.
  10. Methods. Lines 89-100. I suggest to either use a table or to introduce the data within a usual sentence. I have modified this sentence in the manuscript.
  11. Please provide the details of assessments concerning “anti-TPO, anti-TG or TRAB antibodies” and how did you define a positive status. We used the following standards: anty-TPO ( n<35,0 IU/ml), anty Tg: (n<4.11 IU/ml), TRAB (n< 1.8 IU/ml)
  12. Please define vitamin D deficiency (in terms of cut offs). Vitamin D deficiency was diagnosed when the concentration of the 25-hydroxyvitamin Din the blood were lower than <30 nmol/L.

I added it into the manuscript.

  1. Table 1 and 2. Please use left alignment for the first column. Ok, I did it
  2. Legend for Table 3. What do you mean for “risk factors versus…”? Should be risk factors for.. I have changed.
  3. Results. Lines 230-233. A meta-analysis is based on multiple studies, not just one study. Please re-organize the information. Here the meta-analysis ( as I said before – Point 7-) is strictly connected with our analyzed risk factors.  

23. Conclusion -  I suggest to use something other than “inexperience”, maybe defying the number of procedures (since a surgeon that performs 49 thyroidectomies per year  is not inexperienced while the surgeon performing 51 such procedures is “experienced” Ok, It is important suggestion- I agree, according your advise and another reviewer – I changed the conlusion at all.  

Reviewer 2 Report

The manuscript " Risk Factors for calcium-phosphate disorders after thyroid surgery" aims to study the risk factors associated with calcium-phosphate disorders (hypoparathyroidism and hypocalcemia) after thyroidectomy. The manuscript is interesting and clinically relevant. There are some aspects to be considered:

- The abstract contain some abbreviations with the significance (Parathormone PTH)

- The introduction is too objective. some more information about the importance of surgery, the epidemiology of thyroid surgery....

- The methods need improvements:

Please provide the inclusion and exclusion criteria  for the patients.

Some abbreviations (Parathormone- PTH) already appear before in the text;

Please provide the reference or guidelines used for the postoperative hypoparathyroidism and hypocalcemia definition. (Data Collection)

In the results, on table 5, one of the risk factors is the "experienced surgeon". This is very difficult to analyse. Please provide the criteria for this risk factor. The authors only mention number of surgeries. Maybe it should not be considered a risk factor, because it is not objective.

Minor editing of English language required

Author Response

Dear Reviewer.

Thank you for your insightful and valuable comments on the manuscript. Below are our answers. We tried our best to improve the manuscript.

Kind regards,

Beata Wojtczak and Monika Sępek with the authors.

The manuscript " Risk Factors for calcium-phosphate disorders after thyroid surgery" aims to study the risk factors associated with calcium-phosphate disorders (hypoparathyroidism and hypocalcemia) after thyroidectomy. The manuscript is interesting and clinically relevant. There are some aspects to be considered:

- The abstract contain some abbreviations with the significance (Parathormone PTH)

Ok, it was correct.

- The introduction is too objective. some more information about the importance of surgery, the epidemiology of thyroid surgery....

Of course, the introduction was expanded as all the suggestions.

- The methods need improvements:

Please provide the inclusion and exclusion criteria  for the patients.

Ok, we provided inclusion and exclusion criteria for the patients.

Some abbreviations (Parathormone- PTH) already appear before in the text;

It was corrected.

Please provide the reference or guidelines used for the postoperative hypoparathyroidism and hypocalcemia definition. (Data Collection)

---according the 2017 American Thyroid Society guidelines – it was added in the text--

In the results, on table 5, one of the risk factors is the "experienced surgeon". This is very difficult to analyse. Please provide the criteria for this risk factor. The authors only mention number of surgeries. Maybe it should not be considered a risk factor, because it is not objective.

Of course, it is always problem how to define experience or inexperience surgeon. I agree that there is not one proper definition. I agree it is sound “bad inexperienced”, so in all places I gave the number of operation performed.

Comments on the Quality of English Language - Minor editing of English language required

We have done minor English language changed.

Thank you

Beata Wojtczak and Monika Sępek with the authors

Reviewer 3 Report

good work

Author Response

Dear Reviewer.

Thank you for your comments on the manuscript.

Kind regards,

Beata Wojtczak and Monika Sępek with the authors.

Reviewer 4 Report

1. Lines 90-98 - it's better to make a numbered list.

2. In Table 6, line Calcium levels before surgery is the value of the risk ratio 321.378 0.765 135068.248 - is this not a typo? Have the input data been checked for statistical screening for outliers? what method? Table 4 also shows high values for the same indicator, but not to the same extent. We need to discuss the text.

3. Why are patients with repeat thyroid surgery included? It is clear that the risk increases with each operation, but this is not shown statistically by the authors, because. there are too few patients in the secondary group. I would exclude this group.

Author Response

Dear Reviewer.

Thank you for your insightful and valuable comments on the manuscript. Below are our answers. We tried our best to improve the manuscript.

Kind regards,

Beata Wojtczak and Monika Sępek with the authors.

1.Lines 90-98 - it's better to make a numbered list.

Ok, it was done – it is numbered.

2.In Table 6, line Calcium levels before surgery is the value of the risk ratio 321.378 0.765 135068.248 - is this not a typo? Have the input data been checked for statistical screening for outliers? what method? Table 4 also shows high values for the same indicator, but not to the same extent. We need to discuss the text.

Of course, it was mistake , it was corrected.

3.Why are patients with repeat thyroid surgery included? It is clear that the risk increases with each operation, but this is not shown statistically by the authors, because. there are too few patients in the secondary group. I would exclude this group.

Ok, I agree it was a small group, but it was one of the risk factor that is why we take this risk factor. I will comment it in discussion. Now it is difficult to exclude this group, becous it would changed previous statistical analyses. I did the comments in discussion in part about limitations of the study.

Thank you,

Beata Wojtczak and Monika Sępek with the authors

Reviewer 5 Report

Thank you very much for allowing me to review the article entitled "Risk factors for calcium-phosphate disorders after thyroid surgery" (biomedicines-2493542). This work is submitted to the "Molecular and Translational Medicine" section of the Special Issue "Thyroid Disease: From Mechanism to Therapeutic Approaches."

The aim was to analyse risk factors for calcium-phosphate disorders after thyroidectomy: postoperative hypoparathyroidism and hypocalcaemia separately in the postoperative period following thyroid surgery.

The introduction should be expanded, stating the research hypothesis to reach the objective. Since the topic addressed is a problem associated with thyroid surgery, the underlying hypothesis, scientific basis, and clinical applicability should be further developed.

The study was conducted retrospectively on 211 patients who underwent surgery between 2018 and 2020. Therefore, a retrospective case series design was used. The study design should be defined, as the results present odds ratios (OR), indicating the use of a case-control design. However, this design is not described in the materials and methods, and the case criteria and control criteria should be defined. Additionally, the calculation of OR should be incorporated into the statistical analysis.

In the Results section, it is stated that 27 out of the 211 patients studied developed hypothyroidism. It should be indicated that this represents an incidence of 12.79 cases per 100 patients operated on. In Table 2, a comparison of patient characteristics should be provided rather than solely describing the percentages. Table 3 is very interesting, but the font size appears too small, making it difficult to read. Furthermore, comparisons of the presented proportions should be included. It is unnecessary to state "Yes" and "No" since they are complementary results; presenting only "Yes" would be sufficient.

The results of the OR analysis (Tables 4 and 5) should be presented before discussing the principal component analysis. Adjusted ORs should also be calculated to assess the interaction of the different factors studied. ORs should be presented with two decimal places, as using three decimal places can be speculative.

The chi-square test does not need to be mentioned; only the p-value should be reported. The p-value should be presented with three decimal places throughout the study, as is customary. Including more decimal places is unnecessary.

The discussion is very interesting, but the contributions and limitations of the study should be clearly stated, as well as its clinical applicability and potential implications.

The conclusion is repeated at the end of the discussion in the Conclusions section. The conclusion should serve as a summary of the study rather than an additional contribution.

Author Response

Dear Reviewer.

Thank you very much for your insightful and valuable comments on the manuscript. Below are our answers. We tried our best to improve the manuscript.

Kind regards,

Beata Wojtczak and Monika Sępek with the authors.

Thank you very much for allowing me to review the article entitled "Risk factors for calcium-phosphate disorders after thyroid surgery" (biomedicines-2493542). This work is submitted to the "Molecular and Translational Medicine" section of the Special Issue "Thyroid Disease: From Mechanism to Therapeutic Approaches."

The aim was to analyse risk factors for calcium-phosphate disorders after thyroidectomy: postoperative hypoparathyroidism and hypocalcaemia separately in the postoperative period following thyroid surgery.

The introduction should be expanded, stating the research hypothesis to reach the objective. Since the topic addressed is a problem associated with thyroid surgery, the underlying hypothesis, scientific basis, and clinical applicability should be further developed.

Of course, the introduction was expanded as all the suggestions.

The study was conducted retrospectively on 211 patients who underwent surgery between 2018 and 2020. Therefore, a retrospective case series design was used. The study design should be defined, as the results present odds ratios (OR), indicating the use of a case-control design. However, this design is not described in the materials and methods, and the case criteria and control criteria should be defined. Additionally, the calculation of OR should be incorporated into the statistical analysis.

The material was collected retrospectively, so the power analysis of the test was not justified, there was no study design stage, only the material was collected for the purposes of the doctoral dissertation backwards.

In the Results section, it is stated that 27 out of the 211 patients studied developed hypothyroidism. It should be indicated that this represents an incidence of 12.79 cases per 100 patients operated on.

I added this information in the manuscript.  

In Table 2, a comparison of patient characteristics should be provided rather than solely describing the percentages.

The name of the Table 2 was changed because was not correct: Hypoparathyroidism vs. hypocalcemia after thyroid surgery. Characteristic of the patients with hypoparathyroidism and hypocalcemia following thyroidectomy.  The purpose of this Table 2 was not to compare the groups of patients with hypoparathyroidism vs. hypocalcemia, but to describe the group of patients with hypoparathyroidis and separately with hypocalcemia.

Table 3 is very interesting, but the font size appears too small, making it difficult to read.

I will ask the editors to increase the font size, because it is on their side that the final arrangement of the text is made.

Furthermore, comparisons of the presented proportions should be included.

It was done in Table 4, 5 and 6,7.

 It is unnecessary to state "Yes" and "No" since they are complementary results; presenting only "Yes" would be sufficient.

The results of the OR analysis (Tables 4 and 5) should be presented before discussing the principal component analysis. Adjusted ORs should also be calculated to assess the interaction of the different factors studied. ORs should be presented with two decimal places, as using three decimal places can be speculative.

Now the results of the OR analysis (Tables 4 and 5) was presented before discussing the principal component analysis. And the text connected with the principal component analysis was presented following the Table. 4.5. In Tables 4,5,6,7 all the results of the OR analysis was presented with two decimal places and in the manuscript as well.

The chi-square test does not need to be mentioned; only the p-value should be reported. The p-value should be presented with three decimal places throughout the study, as is customary. Including more decimal places is unnecessary.

Of course, we crossed out the chi-square from tables. Now the p-value is presented with three decimal places.

The discussion is very interesting, but the contributions and limitations of the study should be clearly stated, as well as its clinical applicability and potential implications.

Of course, all suggested aspects was added into the discussion.

The conclusion is repeated at the end of the discussion in the Conclusions section. The conclusion should serve as a summary of the study rather than an additional contribution.

Of course, I have been redrafted this part of manuscript to avoid repetition of information in discussions and conclusions.

Round 2

Reviewer 4 Report

The authors responded to the comments of the reviewer. I think that in its present form the article can be recommended for publication.

Author Response

Thank you for your reccomendation. 

Authors of the manuscript.

Reviewer 5 Report

I have carefully reviewed the revised version of the manuscript titled "Risk factors for calcium-phosphate disorders after thyroid surgery" (biomedicines-2493542), along with the authors' response to the suggestions made.

I must say that the manuscript has shown significant improvement. However, I noticed that the authors have not addressed the issue of simplifying the tables where the presence and absence of factors (yes/no) are presented. Since these variables are dichotomous, it is unnecessary to include both options in the table. This modification still needs to be made.

Furthermore, there is a more significant concern that I have regarding the study design. The authors state that the study was conducted retrospectively on 211 patients who underwent surgery between 2018 and 2020, and they used a retrospective case series design. However, the results present odds ratios (OR), indicating the use of a case-control design. Unfortunately, the materials and methods section does not describe this case-control design, and the criteria for defining cases and controls are missing. Additionally, the calculation of OR should be incorporated into the statistical analysis.

The authors' response that the material was collected retrospectively for the purpose of a doctoral dissertation, and there was no study design stage, does not fully address the issue. When presenting OR assessment, it is crucial to define the design used, which, in this case, would be a retrospective case-control design. Moreover, it is essential to specify the reference level for each factor studied in table four. This omission is a serious flaw in the work and should be corrected.

In conclusion, the manuscript has made notable improvements, but the issues related to the tables' presentation and the study design must be addressed and rectified to ensure the validity and clarity of the findings.

Author Response

Dear Reviewer,

Once again, thank you very much for the thorough analysis of our manuscript. Thanks for all the advice and corrections. We analyzed our work again with statistics. Here are our fixes and answers.

Best, Beata Wojtczak

I have carefully reviewed the revised version of the manuscript titled "Risk factors for calcium-phosphate disorders after thyroid surgery" (biomedicines-2493542), along with the authors' response to the suggestions made.

I must say that the manuscript has shown significant improvement. However, I noticed that the authors have not addressed the issue of simplifying the tables where the presence and absence of factors (yes/no) are presented. Since these variables are dichotomous, it is unnecessary to include both options in the table. This modification still needs to be made. We made this modification in Table 3. ( It is in green in the manuscript)

Furthermore, there is a more significant concern that I have regarding the study design. The authors state that the study was conducted retrospectively on 211 patients who underwent surgery between 2018 and 2020, and they used a retrospective case series design. However, the results present odds ratios (OR), indicating the use of a case-control design. Unfortunately, the materials and methods section does not describe this case-control design, and the criteria for defining cases and controls are missing. Additionally, the calculation of OR should be incorporated into the statistical analysis.

The authors' response that the material was collected retrospectively for the purpose of a doctoral dissertation, and there was no study design stage, does not fully address the issue. When presenting OR assessment, it is crucial to define the design used, which, in this case, would be a retrospective case-control design. Moreover, it is essential to specify the reference level for each factor studied in table four. This omission is a serious flaw in the work and should be corrected.

In Tab 4, the adjusted OR values were determined by logistic regression analysis, in which the dependent variable was continuous (it was on a quotient scale - it was a real number). This analysis is continuous. In addition, the criteria (reference level for each factor in Tab 4) are the same as those presented in Tab 3, and in order not to repeat it, it is included in Tab 3. It was not possible to build a statistically significant multi-factor logistic regression model that would take into account the OR assessment interactions between variables and it was comment in discussion.

In conclusion, the manuscript has made notable improvements, but the issues related to the tables' presentation and the study design must be addressed and rectified to ensure the validity and clarity of the findings.

Round 3

Reviewer 5 Report

I have revised again the manuscript entitled "Risk factors for calcium-phosphate disorders after thyroid surgery" (biomedicines-2493542), along with the authors' response to the suggestions made.

I believe the authors have improved the presentation of their work. However, they still do not describe in the material and methods, specifically in the statistical analysis section in tables 4 and 5, the tests presented and the OR presented, so the manuscript is not complete. Furthermore, if the OR is a continuous variable, the units should be indicated both in the material and methods and in the tables.

Author Response

Dear Reviewer,

Thank you for your valuable comment. Now I understand what should be correct/ improved. I did my best changes in tha partaboutr statistical analysis.

This part I have added in the Manuscript.

In the case of the statistical significance ( p<0.05 ) of the test chi2 evaluating the relationship between the

dichotomous variables listed in tables 5 and 7, in order to visualize the results obtained, OR odds ratios were additionally determined. The values ​​were calculated using the standard method based on the constructed 2x2 two-way tables: Tab={[a,b], [c,d]} V (a/c)/(b/d)

Statistical significance of correlations between dichotomous variables and continuous variables presented in Tables 4 and 6 were visualized by determining odds ratios with 95% significance intervals ± 95% CI In this case, the values ​​were estimated using one-way logistic regression. Those for which the range ± 95% CI does not include 1 were considered statistically significant.